# Intra-Host Evolution of SARS-CoV-2 During Persistent Infection of Pediatric COVID-19 Patients

**DOI:** 10.3390/v17101313

**Published:** 2025-09-28

**Authors:** Charlie R. Boyle, Tien Doan, Estefany Rios-Guzman, Jessica Maciuch, Lacy M. Simons, Dulce S. Garcia, David B. Williams, Arghavan Alisoltani, Egon A. Ozer, Ramon Lorenzo-Redondo, Judd F. Hultquist

**Affiliations:** 1Division of Pediatric Infectious Diseases, Anne and Robert H. Lurie Children’s Hospital, Chicago, IL 60611, USA; cboyle@luriechildrens.org; 2Center for Pathogen Genomics and Microbial Evolution, Northwestern University Havey Institute for Global Health, Chicago, IL 60611, USA; buihanhtien.doan@northwestern.edu (T.D.); estefany.guzman@northwestern.edu (E.R.-G.); lacy.simons@northwestern.edu (L.M.S.); dulce.garcia@northwestern.edu (D.S.G.); e-ozer@northwestern.edu (E.A.O.); ramon.lorenzo@northwestern.edu (R.L.-R.); 3Division of Infectious Diseases, Feinberg School of Medicine, Northwestern University, Chicago, IL 60611, USA; 4Division of Allergy and Immunology, Feinberg School of Medicine, Northwestern University, Chicago, IL 60611, USA; jessica.maciuch@northwestern.edu; 5Stanley Manne Children’s Research Institute, Ann & Robert H. Lurie Children’s Hospital of Chicago, Northwestern University Feinberg School of Medicine, Chicago, IL 60611, USA

**Keywords:** Coronavirus disease 2019, COVID-19, severe acute respiratory syndrome coronavirus, SARS-CoV-2, viral evolution, longitudinal cohort, pediatric, intra-host diversity

## Abstract

The Coronavirus disease 2019 (COVID-19) pandemic had a profound global impact, yet children exhibited distinct clinical and epidemiological patterns compared to adults. Pediatric cases of severe acute respiratory syndrome coronavirus 2 (SARS-CoV-2) were generally characterized by milder disease, lower hospitalization rates, and few long-term sequelae. However, a subset of children developed severe complications such as multisystem inflammatory syndrome in children (MIS-C), highlighting the heterogeneity in disease presentation. Differences in immune system maturity and comorbidities likely contribute to the age-dependent manifestation of SARS-CoV-2 and other respiratory viruses. Persistent SARS-CoV-2 infection, particularly in immunocompromised individuals, has been implicated in the emergence of new viral variants with immune escape characteristics due to ongoing viral replication in the presence of selective pressure. While SARS-CoV-2 evolution in persistently infected adults has been well-documented, it is less clear how the virus evolves during persistent infection in the pediatric population. To address this question, we performed viral whole genome sequencing of longitudinal specimens collected from immunocompetent and immunocompromised pediatric COVID-19 patients. Similarly to what has been observed in adult cohorts, mutations associated with enhanced viral fitness and immune escape arose intra-host over time. Intra-host diversity accumulated at similar rates in immunocompetent and immunocompromised children, though more mutations overall were observed in the immunocompromised cohort due to the longer infection time courses. Overall, we identified similar viral evolutionary trends over the course of infection despite clinical differences in pediatric COVID-19 manifestation and severity. This similarity suggests that persistent infection in children may be an additional, but not unique, source of ongoing viral diversification.

## 1. Introduction

The Coronavirus disease 2019 (COVID-19) pandemic had profound global socioeconomical and cultural impacts, but its overall clinical impact on pediatric populations was muted in comparison to adults [1,2]. Most severe acute respiratory syndrome coronavirus 2 (SARS-CoV-2) infections in children are asymptomatic to mild, though more severe sequelae may develop in children with underlying health conditions [1,2,3,4,5,6,7,8,9]. In rare cases, pediatric COVID-19 patients may develop multisystem inflammatory syndrome in children (MIS-C), which can ultimately lead to cardiovascular compromise and shock [10,11]. Nonetheless, clinical metrics of disease severity, hospitalization rates, admission to the intensive care unit (ICU), and rates of long-term sequalae are all decreased in pediatric relative to adult populations [1,6,7]. This pattern of milder disease in children is not unique to SARS-CoV-2, as several respiratory pathogens are known to manifest differently in children verses adults [12,13]. These differences in disease manifestation are thought to be driven by underlying factors such as prior exposure, immune system maturity, and the frequency and nature of other underlying health conditions [14,15,16,17,18,19,20].

Immunocompromised or immunosuppressed people are at particularly high risk for developing severe COVID-19 due to an inability to clear the infection [14,18,19,20,21]. SARS-CoV-2 can persist in these patients for several months, enabling the intra-host evolution of viral variants with resistance to therapeutic pressures. Longitudinal sequencing of SARS-CoV-2 in adult COVID-19 patients with persistent infection has revealed the intra-host emergence of several Spike mutations that confer resistance to neutralizing antibodies and monoclonal antibody therapeutics, such as E340K and E484K [21,22,23,24,25,26,27]. Several of these mutations are shared with those that emerged in assorted variants of concern (VOCs), leading to the theory that these VOCs may have originally evolved in immunocompromised patients with persistent infection [14,16,17,18,19,25,28,29,30,31]. More recently, longitudinal sequencing in immunocompromised patients has identified the intra-host emergence of resistance mutations to administered antiviral therapies [32,33,34,35], though these have not yet emerged in widely circulating strains in the community.

While several studies have explored the evolution of SARS-CoV-2 in adults with persistent COVID-19, much less is known about how the virus evolves and diversifies in a pediatric population. We hypothesized that the unique immune environment of younger pediatric patients would enable greater viral diversification compared to teenagers and adults with fewer differences observed between immunocompromised and immunocompetent individuals. In this single-center, retrospective cohort study, we performed longitudinal viral whole genome sequencing in serially collected specimens from 16 pediatric patients with COVID-19. We assessed viral divergence and diversification over time and in response to administered therapeutics. Similarly to findings in adult cohorts, we observed the intra-host emergence of mutations associated with enhanced viral fitness and immune escape over time. Although rates of intra-host diversity accumulation appeared comparable between immunocompetent and immunocompromised children, the latter group exhibited a greater overall number of mutations, likely reflecting their prolonged infection courses. Taken together, these results suggest that, despite the distinct clinical manifestations and generally milder disease course of pediatric COVID-19, viral evolution within children follows patterns consistent with those seen in adults. Thus, persistent infection in pediatric populations may represent an additional, though not unique, contributor to ongoing SARS-CoV-2 diversification.

## 2. Materials and Methods

### 2.1. Cohort Description

This retrospective cohort study was conducted at Northwestern University’s Feinberg School of Medicine (FSM) in collaboration with Ann & Robert H. Lurie Children’s Hospital of Chicago (LCH) with approval from the LCH Institutional Review Board (study protocols 2020-3792 and 2025-7507). A waiver of informed consent was granted by the IRB. All residual diagnostic specimens testing positive for SARS-CoV-2 at LCH between 22 March 2020, and 22 March 2023 were de-identified and stored in a biobank at the Center for Pathogen Genomics and Microbial Evolution at FSM (n = 11,783 specimens). Patients younger than 20 years of age who provided at least 3 specimens with no more than a 90-day gap between samples were considered for inclusion in the study. We excluded any patients that did not meet the aforementioned requirements. All specimens included in this study were nasopharyngeal swabs collected in viral transport media.

### 2.2. Study Data and Definitions

Through manual chart review, specimens were retrospectively matched with demographic and clinical datasets, including information on sex, age, race, ethnicity, hospitalizations, comorbidities, vaccination history, and laboratory values. Vaccine records for the state of Illinois, including SARS-CoV-2 vaccines, were made available through the Illinois Comprehensive Automated Immunization Registry Exchange (I-CARE). Full vaccination status was defined as 14 days after the second dose of Pfizer-BioNTech (Pfizer) or Moderna mRNA vaccine (Moderna). Patients were categorized as partially vaccinated if they only had a single dose of either the Pfizer or Moderna vaccine. The clinical data included comorbid conditions [defined as cardiac disease, respiratory disease, diabetes, asthma, and obesity (defined as body mass index >95th percentile)], hospitalization and/or ICU admission for COVID-19, presence of viral co-infection, and receipt of therapies for COVID-19 (e.g., hydroxychloroquine, remdesivir, tocilizumab, sotrovimab, dexamethasone, and casirivimab/imdevimab). Patients were considered to be immunocompromised if: they were actively receiving or recently received chemotherapy within the past 30 days (e.g., antithymoglobulin, busulfan, fludarabine, or cyclophosphamide); they had a hematologic malignancy, solid organ transplant, or hematopoietic stem cell transplant; or if they received treatment with tacrolimus, sirolimus, or mycophenolate. Additional data, including absolute lymphocyte count (ALC), was collected to confirm immune suppression. Patients were considered asymptomatic if the clinician noted no symptoms or change from clinical baseline.

### 2.3. Viral RNA Extraction

Residual diagnostic specimens underwent viral RNA extraction using the QIAamp 96 Virus QIAcube HT Kit (Qiagen, Hilden, Germany, cat. no. 57731). SARS-CoV-2 viral load was quantified by quantitative reverse transcription PCR (qRT-PCR) with the CDC 2019-nCoV RT-PCR Diagnostic Panel (CDC, Atlanta, GA, USA, cat 2019-nCoVEUA-01), employing the N1 probe set for SARS-CoV-2 and human RP probes for quality control as previously described (CDC, Atlanta, GA, USA, cat 2019-nCoVEUA-01) [36]. Specimens failing to amplify the RP housekeeping gene were excluded, and those with an N1 probe cycle threshold (Ct) value less than or equal to 35 were considered positive and advanced to whole genome sequencing.

### 2.4. cDNA Synthesis and Viral Genome Amplification

cDNA was synthesized from 11 μL of extracted viral nucleic acids using SuperScript IV (ThermoFisher, Waltham, MA, USA, cat. no. 18090200) with random hexamers according to manufacturer’s specifications. Viral genomic cDNA was then amplified in 2 multiplexed PCR reactions to generate ~400 base pair amplicons tiled across the genome. The multiplex primer set, consisting of two non-overlapping primer pools designed with Primal Scheme, was obtained from Integrated DNA Technologies (IDT) as the Artic V5.3.2 NCOV-2019 Panel (IDT, Coralville, IA, USA, cat. no. 10016495). PCR amplification was performed using Q5 HF DNA Polymerase (NEB, Ipswich, MA, USA, cat. no. M0491L) with 5 μL of cDNA in a 25 μL reaction volume. The thermal cycling program consisted of an initial step of 98 °C for 30 s, then 40 cycles of 98 °C for 15 s and 64 °C for 5 min. Separate reactions were run for each primer pool, with all pre- and post-PCR steps conducted in physically distinct spaces to minimize contamination. Amplicon sets generated for each genome were then pooled prior to sequencing library preparation.

### 2.5. Illumina Sequencing, Genome Assembly, and Variant Calling

Sequencing libraries were prepared from genome amplicon pools using the SeqWell ExpressPlex 384 kit per manufacturer’s instructions (Beverly, MA, USA, product #101221). Pooled libraries of up to 96 genomes were sequenced on the Illumina MiSeq using the V2 500 cycle kit (Illumina, San Diego, CA, USA, product #MS-102-2003). Raw reads were processed with Trimmomatic v0.39 to remove adapters and low-quality sequences, then aligned to the Wuhan-Hu-1 reference genome (accession MN908947.3) using bwa v0.7.18. Alignment pileups were generated from the alignment using samtools v1.21, and consensus sequences were called with iVar v1.4.3 using a minimum depth of 10, a minimum base quality score of 20, and a consensus frequency threshold of 0 (i.e., majority base as the consensus). Genomes with less than 90% coverage were excluded from further analysis. Mutational frequencies were calculated using the iVar “variants” function using a minimum threshold of 3%. Only positions with a mutational frequency higher than the mean error rate (*p* ≤ 0.05 by Fisher’s exact test by iVar) were kept for subsequent analyses.

### 2.6. Phylogenetic Analysis

Genome sequences were aligned using MAFFT v7.453. Lineages were assigned using the Pango classification scheme (https://cov-lineages.org/ (accessed on 4 August 2025); PangoLearn version 9/28/2021) and subsequently grouped in variants using scorpio (scorpio v0.3.12) [37]. To examine molecular evolution and phylogenetic relationships among sequences from immunocompetent and immunocompromised patients, we constructed a Maximum Likelihood phylogeny with IQ-Tree v2.0.5 using its ModelFinder function before each analysis to estimate the nucleotide substitution model best-fitted for each dataset by means of Bayesian information criterion (BIC). We assessed the support of the clusters formed by the different lineages both with the Shimodaira–Hasegawa approximate likelihood-ratio test (SH-aLRT) and with ultrafast bootstrap (UFboot) with 1000 replicates each.

### 2.7. Viral Diversity Analysis

Shannon entropy (Sh) was calculated according to the equation Sh = SUM[−(pi) × log2(pi)], where pi represents the mutational frequency of all nucleotides, including the reference nucleotide, at each position for all reads in a sample. Total Sh across the whole genome or each gene was calculated as the sum of all positions within the specified region and was calculated separately per sample. Only nucleotide substitutions, not insertions or deletions, were considered in this analysis.

### 2.8. Statistical Modeling

Statistical analysis was performed in R v4.5.1. Differences in Shannon entropy between immunocompetent and immunocompromised patients over time were tested using a linear mixed effects model using lme4 (v 1.1-37) and lmerTest (v 3.1-3) controlling for Ct value, remdesivir administration, and patient-level random effects. Remdesivir administration was accounted for in the model as a binary indicator variable if the date of a given sample was taken during a course of Remdesivir treatment or within 7 days of the end of the most recent treatment. Log transformations were used for total Shannon entropy for whole genome model to preserve assumptions of linearity and normality of residuals.

Sh values per position were imputed as 0 for positions that did not pass the 3% mutational frequency threshold and/or failed Fisher’s exact test by iVar, but had a depth greater than or equal to the lower bound to avoid group imbalance in modeling. The lower bound for read depth at each position was calculated as 90% of the lowest depth value greater than or equal to 200 reads. Positions with a depth less than the lower bound were removed from the analysis for all participants across all timepoints. *p*-values were corrected for multiple-hypothesis testing using the Benjamini–Hochberg procedure for FDR correction for per-gene model. All plots were generated with ggplot2 v3.5.2.

### 2.9. Code and Data Availability

Code, tools, and parameter settings required to reproduce these results, including our genome assembly pipeline and within-host variant analysis, is available on GitHub (version 3.17.6) at: https://github.com/Sakazakii/Intra-Host-Evolution-of-SARS-CoV-2-during-Persistent-Infection-of-Pediatric-COVID-19-Patients. All viral whole genome consensus sequences have been uploaded to GISAID (GISAID accession numbers provided in Appendix A). The raw reads from each sequencing run have been deposited to the NCBI Sequence Read Archive (SRA accession numbers provided in Appendix A) and have been uploaded as BioProject PRJNA1295507 (BioSample accession numbers provided in Appendix A). The individualized clinical data reported in this study cannot be deposited in a public repository due to IRB constraints, but population-level metrics are provided in Table 1.

## 3. Results

### 3.1. SARS-CoV-2 Whole Genome Sequencing in Longitudinal Specimens from a Cohort of Pediatric COVID-19 Patients

In March of 2020, a biobank of residual diagnostic specimens (primarily nasopharyngeal swabs) was established at the Center for Pathogen Genomics and Microbial Evolution at the Northwestern University Feinberg School of Medicine from patients who tested positive for SARS-CoV-2 by PCR-based clinical testing at Ann & Robert H. Lurie Children’s Hospital of Chicago (LCH) or at one of its affiliated drive-thru testing centers, outpatient clinics, or immediate care clinics. Over a three-year period between 22 March 2020, and 22 March 2023, we collected 11,783 SARS-CoV-2 positive specimens. During this timeframe, 25 patients provided at least 3 unique specimens with no more than a 90-day gap between samples (n = 112 specimens total). These specimens were retrospectively linked to demographic (e.g., sex, age, race, ethnicity, etc.) and clinical (e.g., hospitalization records, co-morbid conditions, vaccination history, clinical lab values, etc.) datasets. 3 of the 25 patients were found to be older than 20 years of age and excluded from further analysis, leaving 99 specimens collected from 22 patients.

RNA was extracted from each specimen and SARS-CoV-2 viral RNA was measured by quantitative reverse transcription polymerase chain reaction (qRT-PCR) (SARS-CoV-2 N1 primer set, CDC) [36]. Specimens with sufficient viral load [cycle threshold (Ct) value < 35] underwent SARS-CoV-2 whole genome sequencing on the Illumina platform using a multiplex PCR amplicon-based strategy per the ARTIC protocol. We assembled near-complete whole genome sequences from 55 specimens (at least 90% coverage, minimum read depth of 10), 5 of which were unique to a patient (i.e., there were no other sequences available) and were dropped from subsequent analyses. This left a total of 50 whole genome sequences longitudinally collected from 16 patients (consensus sequences and sequence alignments deposited in GISAID and the NCBI Sequence Read Archive, respectively; accession numbers available in Appendix A).

While we initially sought to evaluate differences in SARS-CoV-2 evolution in different age groups, our cohort was too small to achieve sufficient statistical power. We were, however, able to compare viral divergence and diversification in immunocompromised versus immunocompetent patients for comparison to the adult literature. Of the 16 patients with longitudinal sequence information, 11 were immunosuppressed or immunocompromised (defined by the patient’s medical record of having a hematologic malignancy, solid organ transplant, hematopoietic stem cell transplant, or having had treatment with tacrolimus, sirolimus, mycophenolate, thymoglobulin, busulfan, fludarabine, or cyclophosphamide) (Table 1; list of immunocompromising conditions in Table 2). The two groups had similar age, sex, and race/ethnicity distributions. All the immunocompetent and 82% of the immunocompromised patients were not vaccinated at the time of the study, which largely predated vaccine rollout in the pediatric population. Only 1 (20%) of the immunocompetent patients received treatment for COVID-19 (including remdesivir, hydroxychloroquine, dexamethasone, and monoclonal antibodies) as opposed to 8 (73%) of the immunocompromised patients. Four patients received monoclonal antibodies for the treatment of COVID-19, including one immunocompetent patient (tocilizumab) and three immunocompromised patients (one received sotrovimab and two received casirivimab/imdevimab).

Absolute lymphocyte count (ALC) data extracted from complete blood counts collected within the 5 days prior to COVID-19 diagnosis in the immunocompromised patients confirmed immunosuppression compared to expected age-matched standards [mean (SD) = 1114 (832); data not available for immunocompetent patients] (Table 1). This was confirmed in the ALC data collected within two days after diagnosis, with immunocompetent patients having significantly higher ALCs (mean of 3075 compared to 905, respectively). Note there was a relative, but not statistically significant, reduction in the mean ALC after COVID-19 diagnosis in the immunocompromised patient population.

A majority of the immunocompetent patients were asymptomatic (20%) or outpatients (40%) while a majority of the immunocompromised patients were admitted to the hospital (28%), required ICU care (9%), or succumbed to their infection (18%) (Table 1). Despite these differences in outcome, disease severity as measured by the maximally recorded WHO severity score during their infection course was not significantly different between these two groups (mean of 2.4 and 2.2 for immunocompetent and immunocompromised patients, respectively). For reference, a WHO severity score of 2 corresponds to symptomatic infection requiring only ambulatory care (i.e., without the need for supplemental oxygen or hospitalization). This highlights that, on average, both groups experienced relatively mild disease despite the observed differences in clinical care and outcomes.

For the hospitalized patients, the mean length of stay was longer for the immunocompromised patients (114 days) compared to the immunocompetent patients (46.5 days) (Table 1). This generally correlated with longer time courses over which these patients provided SARS-CoV-2 positive specimens (Figure 1A). Patient specimens were not collected over uniform schedules as testing was dictated by the ordering clinician and the residual specimen was collected retrospectively. As some of the patients were asymptomatic, all collection dates are shown relative to the date of first diagnosis. The total length of specimen collection from individual patients ranged from less than a week to over 200 days. The average length of collection was longer in immunocompromised (48 days) compared to immunocompetent patients (27 days), which is consistent with longer persistence in immunocompromised patients as reported in adult populations [21,37].

As described above, Ct values for SARS-CoV-2 viral RNA in each specimen were calculated using a previously validated qRT-PCR assay as a proxy for a viral load (Figure 1B). In general, viral load decreased over time in each patient with some stochasticity. A handful of immunocompromised patients retained higher viral loads (Ct value less than 30) beyond 25 days since initial diagnosis, though this did not clearly correlate with outcome (1 asymptomatic patient, 2 inpatients not requiring ICU care, and 1 patient who died). One immunocompetent outpatient registered a high viral load (low Ct value) 68 days after initial diagnosis, though this was later found to be a reinfection (refer to next section).

### 3.2. Phylogenetic Analysis Reveals Minor Changes in Viral Consensus Sequences over Time

To better characterize the consensus level mutations that arose over time, maximum-likelihood phylogenetic trees were constructed for the 35 longitudinal isolates collected from the 11 immunocompromised patients (Figure 2A) and the 15 longitudinal isolates collected from the 5 immunocompetent patients (Figure 2B). Nearly all the sequences collected from each patient were closely related and belonged to the same lineage/clade, confirming that they were all collected over the course of the same infection. The one exception to this was an immunocompetent patient from which two specimens with identical consensus sequences mapping to lineage AY.25.1 (clade 21J, Delta variant) were collected two days apart and a third specimen mapping to lineage BA.1 (clade 21K, Omicron variant) was collected 66 days later. This third specimen was excluded from further analysis as a reinfection. Longitudinal specimens from the same patient clustered together in a larger subsampling of SARS-CoV-2 whole genome sequences form the US over the time course of this study and clustered with their expected lineage designations (Appendix A).

**Figure 1 viruses-17-01313-f001:**
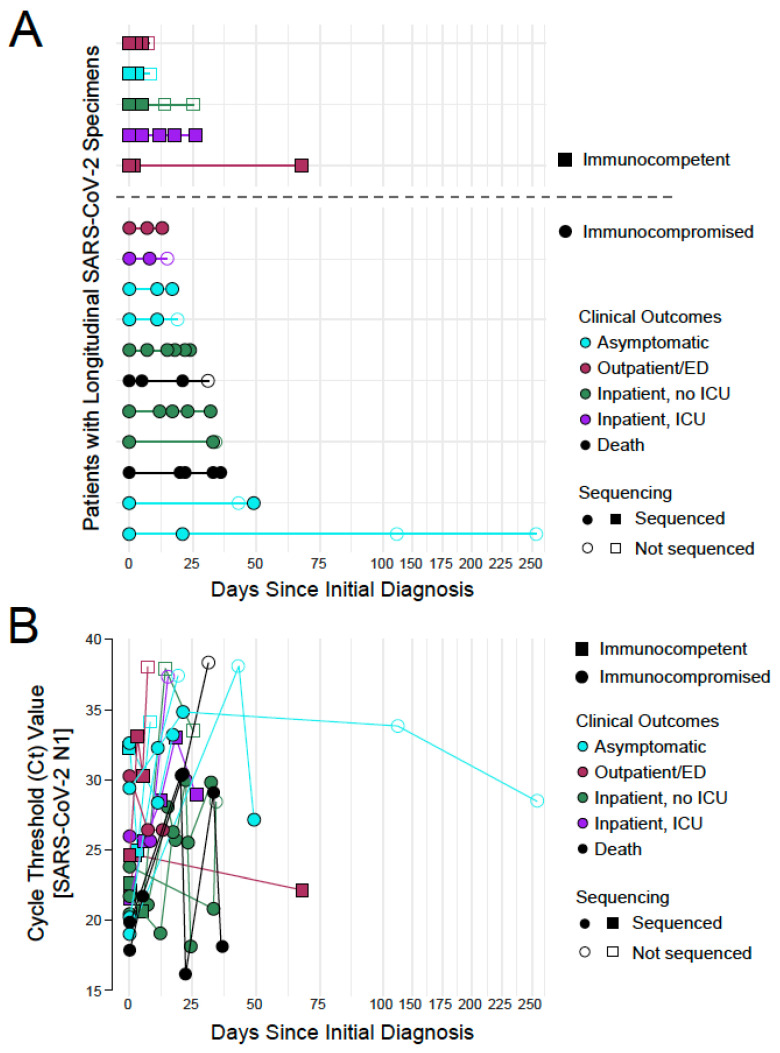
Overview of longitudinal SARS-CoV-2 specimens collected from pediatric patients. (**A**) A graphical representation of the longitudinal specimens included in this study plotted relative to the day of first diagnosis (n = 16 total participants). Participants are grouped by immune status [immunocompetent as squares (n = 5), immunocompromised as circles (n = 11)] and colored by outcome [asymptomatic in blue (n = 5), outpatients in red (n = 3), inpatients never requiring ICU care in green (n = 4), inpatients requiring ICU care in purple (n = 2), and COVID-related deaths in black (n = 2)]. Specimens that yielded a SARS-CoV-2 whole genome sequence are in closed symbols and those that did not are in open symbols. (**B**) Plot of the qRT-PCR Ct values for SARS-CoV-2 N1 in each specimen plotted relative to the day of first diagnosis and visualized using the same schema as in (**A**).

**Figure 2 viruses-17-01313-f002:**
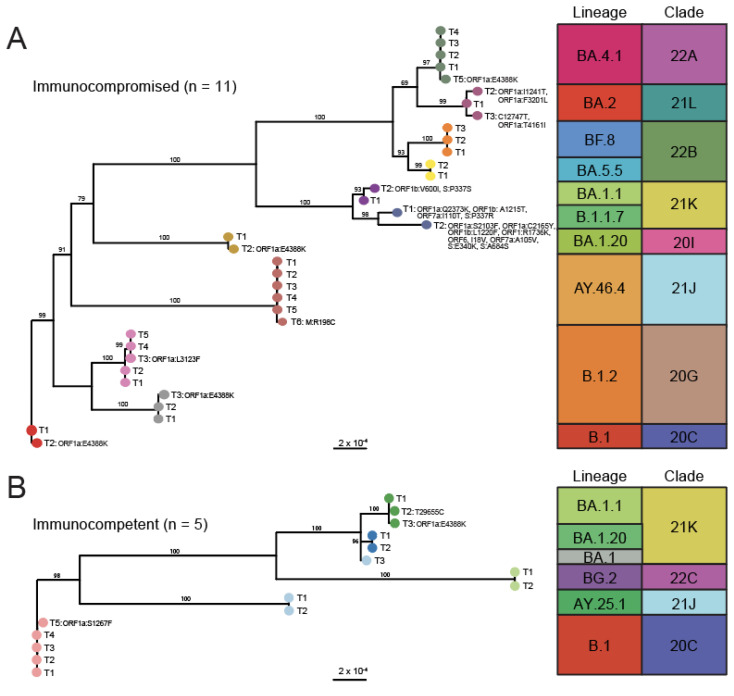
Phylogenetic trees of longitudinally collected SARS-CoV-2 whole genome sequences. Maximum likelihood phylogenetic trees of SARS-CoV-2 whole genome sequences from longitudinally collected clinical specimens from (**A**) immunocompromised (n = 11) and (**B**) immunocompetent (n = 5) patients. Branch tips are colored by patient and labelled by timepoint (T1, T2, etc.). Consensus sequence mutations relative to the nearest branch point are labeled by nucleotide change (synonymous) or open-reading frame and amino acid substitution (non-synonymous). Bootstrap values are given above each branch as a percent of 1000 replicates. The lineage and clade of each sequence is shown on the right.

Over time, the immunocompetent patients saw either no (n = 3, 60%) or minimal (n = 2, 40%) changes in the consensus sequence of the infecting isolate (Figure 2B) with no changes observed in the Spike open reading frame (ORF). On the contrary, only 2 of the 11 (n = 2, 18%) immunocompromised patients had no change in the consensus sequence of the virus over time, with a majority of specimens showing at least one base pair change at later timepoints (Figure 2A). Some patient specimens showed greater divergence, with one set of paired specimens differing by 9 amino acid changes. This included three amino acid substitutions in Spike: P337R (present only at the first timepoint), and E340K and A684S (present at the second timepoint). Notably, this patient was administered sotrovimab prior to collection of the first specimen and variation at positions 337 and 340 in the receptor-binding domain (RBD) have been previously associated with escape from sotrovimab in adult populations [34,38]. A second immunocompromised patient also showed a substitution in Spike at position 337, though to a different amino acid (P337S), and this patient did not receive any monoclonal antibody therapeutics. None of the other patients who received monoclonal antibodies for the treatment of COVID-19 showed any consensus mutations in Spike. Outside of Spike, several patients evolved an ORF1a:E4388K mutation (n = 1 immunocompetent patient and n = 4 immunocompromised patients). While more mutations were observed in the immunocompromised cohort, direct comparison with the immunocompetent cohort is confounded by the larger cohort size and generally greater lengths of time between samplings.

### 3.3. The Rate of Change in SARS-CoV-2 Intra-Host Diversity Does Not Differ by Immune Status in Pediatric Populations

To enable a more granular quantitative comparison of SARS-CoV-2 evolution in the immunocompromised versus immunocompetent pediatric cohorts, we next evaluated intra-host genetic diversity of the viral population in each specimen. Variant allele frequencies per sample were used to calculate mutational frequency and Shannon entropy (Sh) at each position across the genome relative to the Wuhan-Hu-1 reference genome (accession MN908947.3). Only positions with a variant allele frequency greater than or equal to 3% and with a significant *p*-value (*p*-value < 0.05) were included in these analyses. Overlaying all the mutational frequency data from the immunocompetent (left) and immunocompromised (right) patients provides a snapshot of overall positional diversity (Figure 3A). Viruses from both sets of patients have a large number of positions with 100% mutational frequency corresponding with known mutations in the originating lineages. To better visualize this, Sh was calculated as nucleotide-agnostic metric of variability at each position and plotted similarly (Figure 3B, refer to Methods for Sh formula and GitHub for associated code). Diversity was dispersed relatively equally along the genome in both cohorts, though the immunocompromised patients tended to have more sites with higher entropy values (>0.5). Again, however, this is confounded by the longer infection time courses in these patients.

To better visualize the change over time, the Sh per genome position was plotted for each specimen in each patient independently (Appendix A). For this analysis, only positions with a read depth greater than or equal to the lower bound across all timepoints within a patient were included (lower bound calculated as 90% of the lowest depth value exceeding 400 reads). If the read depth failed to cross the lower bound in any one specimen, it was dropped from the analysis across all specimens from that patient. In general, Sh increased over time in each patient with more obvious increases observed with longer times between sampling. To better quantify this, the total Sh for each specimen across all positions was calculated and plotted over time (Figure 4A, refer to calculations in Methods). Linear mixed-effects models were independently fit to the immunocompetent and immunocompromised specimens, which revealed a clear correlation between days since initial diagnosis and entropy. Adjusting for Ct value, remdesivir administration, and patient-level effects, there was no significant difference between the immunocompromised and immunocompetent patients over time, with both accumulating total entropy at very similar rates (*p*-value = 0.401).

While the rate of accumulation was similar, there could be positional differences in where genomic diversity was accumulating in each cohort. To assess this, the total Sh for each viral ORF across all specimens was calculated and compared between the immunocompetent and immunocompromised patients (Appendix A). To ensure direct comparison, the lower bound was calculated as 90% of the smallest depth value within the same position across the dataset that exceeds 200 reads. Positions with a depth less than the lower bound were removed from the analysis for all participants across all timepoints. While some ORFs accumulated more diversity than others (i.e., nsp3, nsp12, Spike, etc.), there were no significant differences between the immunocompetent and immunocompromised groups over time as measured by a linear mixed-effects model adjusting for Ct value, remdesivir administration, and patient-level effects.

While there were no significant differences in intra-host diversity by ORF, it is still possible that specific positions would be preferentially selected for based on immune status. To assess this, we extracted mutational frequency data for all positions in Spike with mutation frequency greater than or equal to 3% (or less than or equal to 97%) as compared to the Wuhan-Hu-1 reference genome (accession MN908947.3) (Appendix A). A modified heatmap of the mutational frequency by position was plotted relative to the Spike domain structure with diversity in each patient represented by a line above (immunocompetent) or below (immunocompromised) the schematic (Figure 4B). If positional diversity was observed at multiple timepoints, the greatest change in diversity was visualized. Non-synonymous changes at 9 sites were found in 2 or more patients (vertical lines and labels). While only the changes at P337 reached consensus (Figure 2A), several other minority variants previously associated with immune escape were observed in multiple patients, including changes at D405 and K417. Nevertheless, these were generally observed in both the immunocompromised and immunocompetent cohorts. Taken together, these data suggest similar patterns of SARS-CoV-2 intra-host evolution between immunocompromised and immunocompetent pediatric COVID-19 patients, though additional sequencing in larger cohorts will be required to validate these results.

## 4. Discussion

This study presents a genome-wide longitudinal analysis of SARS-CoV-2 intra-host evolution in pediatric COVID-19 patients, comparing divergence and intra-host diversity between immunocompetent and immunocompromised children. While prior studies have largely focused on adult cases of persistent infection as sources of SARS-CoV-2 variant emergence, our findings demonstrate that pediatric patients, especially those with impaired immune function, are also capable of sustaining prolonged viral replication and developing genetically divergent viral populations. Consistent with previous studies in adult immunocompromised patients [13,16,30], our pediatric cohort demonstrated prolonged infection in immunocompromised children with sustained viral replication evidenced by persistently low Ct values, extended PCR positivity (up to 256 days), and the accumulation of mutations over time. These findings reinforce the need to view viral persistence in immunocompromised children as a plausible contributor to the ongoing viral genetic landscape.

SARS-CoV-2 accumulated non-synonymous mutations in several immunocompromised patients over time, including several mutations in regions of Spike with known immunologic significance [39,40,41,42]. For example, one immunocompromised patient who received the monoclonal antibody sotrovimab prior to specimen collection acquired two mutations in the Spike RBD known to confer sotrovimab resistance: P337R and E340K. Sotrovimab targets a highly conserved epitope on the RBD and its binding is facilitated by a proteoglycan at residue N343 [43]. Mutations at positions P337 and E340 are exceedingly rare (<0.1% global prevalence) and have been noted by multiple studies as hallmark resistance mutations to sotrovimab, which has been confirmed in a number of neutralization assays pseudovirus neutralization assays when exposed to sotrovimab [44,45,46,47,48,49,50]. It is notable then that a second pediatric patient in this study evolved a P337S mutation despite not having received sotrovimab or any other monoclonal antibody therapeutic. This may suggest a unique selective pressure in this patient that warrants additional surveillance in similar pediatric patients. That being said, sotrovimab is no longer recommended for clinical use following global emergence of the BA.2.86 lineage and its descendants, all of which encode a Spike K356T mutation that ablates sotrovimab binding through addition of a glycosylation site and epitope masking [51]. Furthermore, the three other patients who received monoclonal antibody therapeutics in our cohort did not evolve any resistance mutations. Regardless, the recurrence of these substitutions in our cohort underscores that selective pressures from therapeutic interventions can shape intra-host viral populations in pediatric patients similarly to adults [14,33,52,53].

Outside of Spike, longitudinal sequencing of SARS-CoV-2 in several pediatric patients identified the convergent evolution of an ORF1a:E4388K mutation. This corresponds to an E135K mutation in nsp10, a virally encoded cofactor that binds to and enhances the activities of the nsp14 exoribonuclease and the nsp16 methyltransferase. Position 135 is near the C-terminal tail of nsp10, which was truncated in available co-crystal structures and is not thought to play a direct role in mediating the nsp14 or nsp16 interaction [54,55]. This mutation was exceedingly rare over the timeframe of this study, but has appeared more frequently in some Omicron sublineages, including in the currently circulating lineages LP.8.1 and XFG [56]. If and how this mutation may alter the function of these enzymatic complexes or alter viral fitness has not yet been explored.

While several non-synonymous mutations were observed at the consensus level in our immunocompromised cohort, this was rare in our immunocompetent cohort. That being said, there were fewer immunocompetent patients that fit our inclusion criteria and they generally had shorter infection time courses, as expected. Therefore, to better compare viral evolution in our immunocompromised versus immunocompetent patients, we conducted an analysis of intra-host variation across the genome over time. A majority of observed variants were present at low to moderate frequencies, consistent with previous reports suggesting that intra-host variation in SARS-CoV-2 is typically limited during acute infection [39,40,41]. Nevertheless, this diversity accumulated at similar rates in both cohorts, even when controlling for time, viral load, and patient-to-patient variability. Furthermore, immune status was not significantly associated with total Shannon entropy in any particular ORF, suggesting lack of a localized selective pressure in one cohort versus the other. Focusing on variation observed in Spike, there were some positions that varied in multiple patients over time, including several positions in the Spike RBD known to confer neutralizing antibody escape (i.e., N405D, K417T, etc.) [53,57,58]. However, these were largely observed in both immunocompetent and immunocompromised patients.

Taken together, these data suggest that the rate of SARS-CoV-2 intra-host evolution is the same regardless of a patient’s immune status, but that longer persistence allows longer times for mutations to arise in immunocompromised patients. Prolonged infection then ultimately provides the diversity required to respond to selective pressures when they arise, including to monoclonal antibody therapeutics and other treatment regimens. These data are consistent with what has been previously reported in adult populations [22,29,59].

There are several important limitations to our study. Our study is limited by a small sample size, which decreased our statistical power and prevented us from comparing viral evolution across different age categories. Our inclusion criteria requiring at least 3 longitudinal specimens from each patient made inclusion of immunocompromised individuals more likely as repeat testing of these patients was standard of care to evaluate risk for dissemination of disease, determine length of therapy for remdesivir, and inform additional clinical decision-making practices. Despite the higher number of immunocompromised individuals in the study, there was substantial diversity in the immunosuppressive conditions and treatments administered (Table 2), which prevented a more granular analysis. Understanding how different immunocompromising conditions—such as a particular chemotherapy, underlying immunodeficiency, immunosuppressive therapy, or oncologic malignancy—may drive unique patterns of viral persistence and evolution remains a major knowledge gap. There is some evidence correlating the use of calcineurin inhibitors with increased development of escape mutations within longitudinally sequenced SARS-CoV-2, but more research in this area is sorely needed [46].

Additionally, this study followed a single-site, retrospective design with non-standardized sampling intervals, making the number of samples per patient and overall duration of sample collection variable. While we strived to account for this by incorporating time into our statistical models, it may skew data interpretation. Finally, these studies were conducted with specimens collected prior to 2023 and it is possible that newly emergent variants of SARS-CoV-2 might have different behavioral patterns. To better assess evolution over patient age, future studies should incorporate larger cohorts that include both pediatric and adult populations for direct comparison, ideally with functional assays of emergent mutations to investigate host–virus immune interactions longitudinally.

Despite these limitations, this study fills an important research gap providing whole genome sequencing analysis of longitudinal SARS-CoV-2 isolates from pediatric COVID-19 patients at a quaternary care hospital. Our work highlights that persistent SARS-CoV-2 infection in immunocompromised pediatric patients can drive intra-host viral evolution and the emergence of critical immune escape variants, similar to what has been observed in adult populations. This suggests that persistent infection in children may be an additional, but not unique, source of ongoing viral diversification and highlights the need for viral surveillance efforts to include pediatric patients in their design. Furthermore, it emphasizes the need tailored clinical management strategies for immunocompromised patients across all age groups to limit the emergence and spread of immune escape variants.

## Figures and Tables

**Figure 3 viruses-17-01313-f003:**
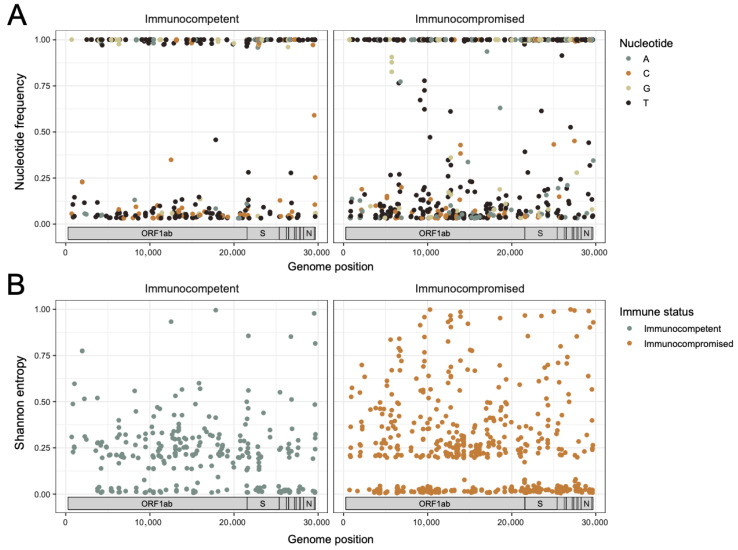
Intra-host diversity across the SARS-CoV-2 genome. (**A**) Dot plots showing mutational frequency per alternative alleles across the genome relative to the Wuhan-Hu-1 reference genome calculated as a fraction of raw reads for immunocompetent (left, n = 15 specimens) and immunocompromised (right, n = 35 specimens) patients, color-coded by nucleotide. The data from all patients and all timepoints per immune status category are overlaid. Only positions with a variable allele frequency greater than or equal to 3% and sequencing depth > 400 reads were included. (**B**) Dot plots showing Shannon entropy per position across the genome for the sites shown in (**A**) from immunocompetent (left, n = 15 specimens) and immunocompromised (right, n = 35 specimens) patients. The data from all patients and all timepoints per immune status category are overlaid.

**Figure 4 viruses-17-01313-f004:**
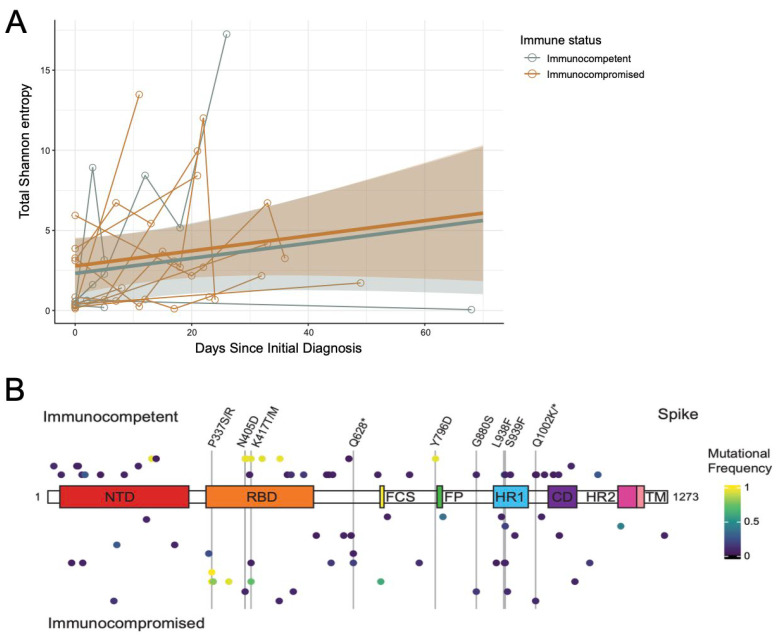
Similar trends in intra-host diversification of SARS-CoV-2 are observed in immunocompromised and immunocompetent pediatric COVID-19 patients. (**A**) Line plot of Shannon entropy [Sh = SUM[−(pi) × log2(pi)], where pi is nucleotide frequency] across the genome of each isolate with longitudinal specimens from the same patient connected by lines. Sh values per position were imputed as 0 for positions that did not pass the 3% mutational frequency threshold and/or failed iVar test (*p* ≤ 0.05 by Fisher’s exact test by iVar) but have a depth greater than or equal to lower bound to avoid group imbalance in modeling. The lower bound was calculated as 90% of the smallest depth value within the same position across the dataset that exceeds 200 reads. A linear mixed-effects model evaluating the trend in total Sh over time in immunocompetent (blue) and immunocompromised (orange) pediatric patients was conducted controlling for Ct, time between sample acquisition, remdesivir administration, and within-patient variability. (**B**) Dot plot of mutational frequency by position across the Spike open reading frame. Diversified sites in one patient are all equidistant from the Spike domain schematic (immunocompetent above, immunocompromised below). Only positional changes that resulted in a non-synonymous change with a variable allele frequency > 3% or <97% and a sequencing depth > 10 reads in at least 1 timepoint are shown. Common sites of diversification between two or more patients are labelled with vertical lines and the amino acid substitution.

**Table 1 viruses-17-01313-t001:** Summary of Demographics, Comorbidities, and Clinical Outcomes of Study Patients.

**Clinical Variables**	**Immunocompetent Patients** **n = 5**	**Immunocompromised Patients ^†^** **n = 11**
Age (years)	6 (7.97)	9.2 (5.65)
Sex (male)	3 (60%)	7 (64%)
Race/Ethnicity		
Non-Hispanic, White	2 (40%)	5 (45%)
Non-Hispanic, Black	0 (0%)	1 (9%)
Non-Hispanic, Other	0 (0%)	0 (0%)
Hispanic	3 (60%)	5 (45%)
BMI	19.95 (8.5)	20.18 (6.1)
Length of Hospitalization (days)	46.5 (55.9)	114 (136)
Patients with Comorbid Conditions ^‡^	2 (40%)	1 (10%)
Number of Patients Who Received Treatment for COVID-19 (any)	1 (20%)	8 (73%)
Remdesivir	0 (0%)	7 (64%)
Dexamethasone	0 (0%)	1 (9%)
Sotrovimab	0 (0%)	1 (9%)
Tocilizumab	1 (20%)	0 (0%)
Casirivimab/imdevimab	0 (0%)	2 (18%)
Hydroxychloroquine	1 (20%)	0 (0%)
Vaccination Status at time of Diagnosis		
Unvaccinated	5 (100%)	8 (73%)
Partially Vaccinated ^◊^	0 (0%)	1 (9%)
Fully Vaccinated *	0 (0%)	2 (18%)
Absolute Lymphocyte Count Prior to COVID-19 Diagnosis	N/A	1114 (832)
Absolute Lymphocyte Count After COVID-19 Diagnosis	3075 (672)	905 (825)
Maximum WHO Severity Score	2.4 (2.79)	2.2 (2.4)
Outcome		
Asymptomatic	1 (20%)	4 (36%)
Outpatient	2 (40%)	1 (9%)
Inpatient, No ICU	1 (20%)	3 (28%)
Inpatient, ICU Admission	1 (20%)	1 (9%)
Death	0 (0%)	2 (18%)
Maximum Respiratory Requirements		
Room Air	4 (80%)	10 (91%)
Nasal Cannula	0 (0%)	0 (0%)
Positive Pressure Ventilation	0 (0%)	0 (0%)
Intubation	1 (20%)	1 (9%)

Data are represented as mean (standard deviation) or number (%) for continuous or categorical variables, respectively. ^†^ Immunocompromising conditions or medications: hematologic malignancy, solid organ transplant, hematopoietic stem cell transplant, tacrolimus, sirolimus, mycophenolate, thymoglobulin, busulfan, fludarabine, cyclophosphamide. ^‡^ Comorbid conditions: cardiac disease, respiratory disease, diabetes, obesity, asthma. ^◊^ Partially Vaccinated: one dose of Moderna or Pfizer mRNA vaccines. * Fully Vaccinated: at least two doses of either Moderna or Pfizer mRNA vaccines.

**Table 2 viruses-17-01313-t002:** Immunocompromising conditions and medications.

**Patient**	**Immunocompromising Conditions/Medications**
1	Osteosarcoma, methotrexate
2	Living donor kidney transplant, mycophenolate mofetil, tacrolimus
3	CAR-T, blinatumomab
4	T-ALL
5	Neuroblastoma, hematopoietic stem cell transplant, thiotepa, cyclophosphamide, eculizumab
6	Living donor kidney transplant, mycophenolate mofetil
7	ALL, CAR-T, hematopoietic stem cell transplant
8	B-ALL, CAR-T, fludarabine, cyclophosphamide
9	Mesenchymal stem cell infusion
10	AML, hematopoietic stem cell transplant, fludarabine, busulfan, antithymocyte globulin
11	B-ALL, hematopoietic stem cell transplant, CAR-T, inotuzimab

## Data Availability

Code, tools, and parameter settings required to reproduce these results, including our genome assembly pipeline and within-host variant analysis, is available on GitHub at: https://github.com/Sakazakii/Intra-Host-Evolution-of-SARS-CoV-2-during-Persistent-Infection-of-Pediatric-COVID-19-Patients. All viral whole genome consensus sequences have been uploaded to GISAID (GISAID accession numbers provided in Appendix A). The raw reads from each sequencing run have been deposited to the NCBI Sequence Read Archive (SRA accession numbers provided in Appendix A) and have been uploaded as BioProject PRJNA1295507 (BioSample accession numbers provided in Appendix A). The individualized clinical data reported in this study cannot be deposited in a public repository due to IRB constraints, but population-level metrics are provided in Table 1.

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
