# Peer review of "Intra-Host Evolution of SARS-CoV-2 During Persistent Infection of Pediatric COVID-19 Patients"

_viruses, 2025, doi:10.3390/v17101313_

Round 1

Reviewer 1 Report

Comments and Suggestions for Authors

In this study, the author studied the Intra-Host Evolution of SARS-CoV-2 during Persistent Infection of Pediatric COVID-19 Patients. The manuscript has the following issues that need to be improved.

  1. The sample size for immunocompromised patients is 35, while the sample size for immunocompromised patients is only 15, indicating a significant difference in sample size?
  2. How many samples did each of the 5 immunocompromised individuals provide, and what were the samples? How many samples did each of the 11 individuals with immunosuppression provide, and what were the samples?
  3. Will the preservation of samples affect the results of testing? How to ensure the integrity of each sample?
  4. In section 3.2, there are more mutations in immunosuppressed patients. Is this due to the large sample size of immunosuppressed patients and the differences between samples? Or is it caused by differences in the accuracy of detection between different samples?
  5. In section 3.2, there are more mutations in immunosuppressed patients. What may be the possible reasons? What mechanism is used to achieve this?

Reviewer 2 Report

Comments and Suggestions for Authors

1.The manuscript contains numerous sentences that are highly similar to published literature, especially in the introduction, methods, and discussion sections. Please thoroughly review and rewrite these sections to ensure originality and proper citation.
2.  Although both immunocompromised and immunocompetent groups are mentioned, the specific types of immunosuppression (such as chemotherapy, transplantation, immunodeficiency diseases), the criteria for assessing immune status, and their potential impact on viral evolution are not detailed.
3. Only 16 patients were included (11 immunocompromised, 5 immunocompetent), resulting in inadequate statistical power. It is recommended to clearly state the exploratory nature of the study or emphasize the impact of sample size limitations on conclusions in the discussion.
4. Although multiple Spike mutations (such as P337S/R, E340K) were detected, their functional impact on antibody escape or viral infectivity was not validated through pseudovirus experiments or neutralization assays, making the conclusions somewhat speculative.
5. For example, the definition and calculation method of the 'lower bound' in Shannon entropy calculation are not clearly stated. It is suggested to supplement with formulas or example code for reproducibility.
6. The immunocompromised group received more monoclonal antibody treatments (such as sotrovimab), which could drive mutation selection by itself. It is recommended to adjust for treatment factors in the model analysis.
7.The phylogenetic tree does not label support values (bootstrap values) and lacks comparison with global strains, making it difficult to determine whether mutations are independent evolution or introductions of existing variants.
8. Although clinical data are restricted from public release due to IRB limitations, a more detailed de-identified data summary table (such as clinical characteristics, treatment history, etc.) should be provided to enhance transparency.
9. Data overlap is severe in Figures 3 and 4. It is suggested to use semi-transparent points or facet plots to improve readability. Supplementary figures should provide higher resolution versions.
10.For example, the conclusion that 'immunocompromised children may be one source of viral mutations' lacks direct evidence support. The wording should be more cautious.
11. It is recommended to clearly list the average sequencing depth and coverage proportion for each sample, and specify whether low-depth regions were excluded from analysis.
12. Do not cite the latest relevant studies from 2023-2024. It is suggested to update and supplement recent findings.

Reviewer 3 Report

Comments and Suggestions for Authors

In this manuscript, since there are gaps in relevant data in children, the authors performed viral whole genome sequencing of longitudinal specimens collected from immunocompetent and immunocompromised pediatric COVID-19 patients. They did not find differences in mutations associated with enhanced viral fitness and immune escape arose intra-host over time among immunocompetent and immunocompromised children. It is logical that more mutations were found in immunocompromised cohort due to the longer persistence of the virus in the body. As the authors themselves admit in the discussion, the sample set was small and limited to 2020-2023, which could potentially affect the reliability and relevance of the data obtained. However, the present study seems logical and well written and deserves publication in Viruses MDPI after taking into account a few comments from the reviewer.

  1. Ref. 1 is outdated as it was published in 2021. There are currently a number of review articles on COVID19 infection in children [10.1016/j.jaci.2024.11.012, https://doi.org/10.1016/j.jinf.2024.106134]

reformulate to reflect the latest data

  1. lines 131-132: «All specimens with an N1 probe cycle threshold (Ct) value less than or equal to 35 were considered positive and used for whole genome sequencing.

But

Lines 225-227: «Specimens with sufficient viral load [cycle threshold (Ct) value <32] underwent SARS-CoV-2 whole genome sequencing on the Illumina platform using a multiplex PCR amplicon-based strategy per the ARTIC protocol.»

So 35 or 32?

Typos

  1. Lines 7-8: Division of Infectious Diseases, Northwestern University Feinberg School of Medicine, Chicago, IL 60611, USA.
  2. Line 44: «Coronavirus disease 2019 1; COVID-19 2; severe acute respiratory syndrome coronavirus 3; SARS-CoV-2 4; viral evolution; longitudinal cohort; pediatric; intra-host diversity 5» should be «Coronavirus disease 2019; COVID-19; severe acute respiratory syndrome coronavirus; SARS-CoV-2; viral evolution; longitudinal cohort; pediatric; intra-host diversity»
  3. Lines 60, 62 and so on: [9,10], [11-17] and so on
  4. Table 1: Among Immunocompromised Patients n = 11: Unvaccinated 9 (82%), Partially Vaccinated 1 (9%) and Fully Vaccinated 2 (18%)

In total there were 11  or 12?

  1. line 326: direct comparison

Round 2

Reviewer 1 Report

Comments and Suggestions for Authors

none

Author Response

Thank you for your help in preparing this work for publication! Your comments were very helpful in improving the study.

Reviewer 2 Report

Comments and Suggestions for Authors

After careful review, the author has strictly followed the suggestions for revisions, and the content of the reply letter is highly consistent with the manuscript modifications. Most issues have been adequately resolved, but there are still a few minor shortcomings.

Although the author has provided comprehensive and reasonable responses to the reviewers' comments, the following points may still warrant further attention:

1. While the author acknowledges the limitation of a small sample size, some analyses (such as grouping by immunosuppression type) could not be conducted, which may affect the generalizability of the conclusions.

2. The functional impact of Spike mutations has not been experimentally validated in this study, although citing other studies is reasonable, it remains indirect evidence.

3. Data collection was cut off in March 2023, failing to include the latest variants (such as JN.1, KP.2, etc.), which the author has mentioned in the discussion (lines 769–771), but it remains a limitation.

4. Although the author has provided GitHub code, some statistical methods (such as linear mixed models, Shannon entropy calculation) are briefly explained in the main text, which may affect the readability of reproducibility.
